# Evolution enhances mutational robustness and suppresses the emergence of a new phenotype: A new computational approach for studying evolution

**Tadamune Kaneko**[1,2☯], **Macoto Kikuchi**[1,2☯]*

**1** Department of Physics, Osaka University, Toyonaka, Japan, **2** Cybermedia Center, Osaka University, Toyonaka, Japan

☯ These authors contributed equally to this work.

\* kikuchi.macoto.cmc@osaka-u.ac.jp

**Data Availability Statement:** Source codes and data are available at Zenodo repository (DOI:10.5281/zenodo.4409496).

## Abstract

The aim of this paper is two-fold. First, we propose a new computational method to investigate the particularities of evolution. Second, we apply this method to a model of gene regulatory networks (GRNs) and explore the evolution of mutational robustness and bistability. Living systems have developed their functions through evolutionary processes. To understand the particularities of this process theoretically, evolutionary simulation (ES) alone is insufficient because the outcomes of ES depend on evolutionary pathways. We need a reference system for comparison. An appropriate reference system for this purpose is an ensemble of the randomly sampled genotypes. However, generating high-fitness genotypes by simple random sampling is difficult because such genotypes are rare. In this study, we used the multicanonical Monte Carlo method developed in statistical physics to construct a reference ensemble of GRNs and compared it with the outcomes of ES. We obtained the following results. First, mutational robustness was significantly higher in ES than in the reference ensemble at the same fitness level. Second, the emergence of a new phenotype, bistability, was delayed in evolution. Third, the bistable group of GRNs contains many mutationally fragile GRNs compared with those in the non-bistable group. This suggests that the delayed emergence of bistability is a consequence of the mutation-selection mechanism.

## Author summary

Living systems are products of evolution, and their present forms reflect their evolutionary history. Thus, to investigate the particularity of the evolutionary process by computer simulations, an appropriate reference system is needed for comparison with the outcomes of evolutionary simulations. In this study, we considered a model of gene regulatory networks (GRNs). Our idea was to construct a reference ensemble comprising randomly generated GRNs. To produce GRNs with high fitness values, which are rare, we employed a "rare event sampling" method developed in statistical physics. In particular, we focused

**Funding:** The authors received no specific funding for this work.

**Competing interests:** The authors have declared that no competing interests exist.

on the evolution of mutational robustness. Living systems do not lose viability readily, even when some genes are mutated. This trait, called mutational robustness, has developed throughout evolution, along with functionality. Using the abovementioned method, we found that mutational robustness resulting from evolution exceeded that of the reference set. Therefore, mutational robustness is enhanced by evolution. We also found that the emergence of a new phenotype was significantly delayed in evolution. Our results suggest that this delay is a consequence of the fact that mutationally robust GRNs are favored by evolution.

## Introduction

As living systems have developed through the long history of evolution, their present forms reflect their evolutionary histories. Thus, some properties of living systems should be consequences of the particularity inherent in evolution. In contrast, some properties may be more commonly observed, so that they do not depend on the evolutionary pathway. In this respect, the common properties and particularities of evolution are of specific interest. However, experimental studies on this topic are limited because the existing living organisms are products of evolution; thus, evolutionary experiments can only provide outcomes that reflect evolutionary histories. Therefore, computational methods are indispensable for obtaining information on evolutionary processes.

Although evolutionary simulation (ES) is a powerful method for studying the evolutionary process, its outcomes depend strongly on evolutionary pathways; therefore, ES alone is insufficient for our purposes and we need a reference system for comparison. In this study, we investigated the evolution of gene regulatory networks (GRNs). The reference system that we consider appropriate for this case is an ensemble of randomly sampled GRNs. If some properties are commonly observed in the ensemble, they should be realized irrespective of the evolutionary pathway. If there are some differences between the results of the ES and reference ensemble, they are manifestations of the particularity of evolution.

The aim of this paper is two-fold. First, we propose a research method for determining the properties of the evolutionary process by comparing a randomly sampled ensemble of genotypes obtained using a method based on statistical mechanics with genotypes obtained using the ES. Second, we apply it to a model of GRNs and investigate the evolution of mutational robustness and emergence of bistability.

Let us discuss our methodology concretely in the context of mutational robustness and bistability of GRNs. Living systems possess many types of robustness including that against environmental and internal noises as well as that against genomic mutations [1–3]. Among the various types of robustness, the most important is mutational robustness, which enables a living system to retain its function and continue to exist despite genome mutation.

Mutational robustness has been demonstrated experimentally. For example, comprehensive single-gene knockout experiments for bacteria and yeasts have revealed that most gene knockouts do not affect viability [4–6]. An artificial rewiring experiment for the GRN of *E. coli* showed that the bacterium remains viable after most artificial additions of regulatory links [7]. Such mutational robustness must have been acquired through an evolutionary process. Mutationally robust genotypes have been selected because of the mutation-selection mechanism, which is not directly related to any function. In this respect, enhancement of mutational robustness through evolution is called "second-order selection" [2].

Random sampling is suitable for identifying properties that do not depend on evolutionary history. Ciliberti *et al.* conducted random sampling of GRNs to investigate mutational robustness [8, 9]. The fitness of their model had only two values—viable and non-viable. They investigated the interrelations of viable GRNs and argued that most GRNs belong to a large cluster connected by neutral mutations, similar to the neutral networks found in the RNA sequence space [10]. However, such a simple random sampling method is not useful for systems with a more complex fitness landscape, because highly fit GRNs are rare. Burda *et al.* and Zagorski *et al.* employed the Markov chain Monte Carlo method to sample highly fit GRNs [11, 12]. They found that GRNs exhibiting multistability contained a common network motif.

The abovementioned methods are insufficient for sampling GRNs with a wide range of fitness levels. Saito and Kikuchi proposed the use of the multicanonical Monte Carlo (McMC) method to investigate the mutational robustness of GRNs [13]. McMC was originally developed in statistical physics for sampling configurations within a wide range of energies [14, 15]. However, this method has also been found to be useful for sampling nonphysical systems [16]. It is particularly effective for generating very rare states and estimating the probabilities of their appearance [16–20].

Nagata and Kikuchi investigated a GRN model using McMC [21]; they regarded fitness as the "energy" of GRNs and sampled GRNs with very low fitness to very high fitness uniformly and randomly. They considered a neural network-like model of GRNs, with one input gene and one output gene, and set the fitness in a manner that fitness was high if the GRN responded sensitively to changes in input. As each gene in their model did not respond ultrasensitively [22, 23], and the network structures were restricted, simple network motifs did not give rise to bistability [24]. Despite this, they found that highly fit GRNs always exhibited bistability. Therefore, bistability has emerged as a consequence of the cooperation of many genes. According to their results, a new phenotype of bistability appears regardless of the evolutionary path. They also found that mutationally robust GRNs were not rare among highly fit GRNs.

Bistable or multistable responses of GRNs are widely observed in living systems. The best-known example is the toggle switch for lysogenic-lytic transition in phage λ, which has been extensively studied both experimentally and theoretically [25–28]. Another example is the cdk1 activation system in *Xenopus* eggs [29–31]. The bistable switches of GRNs are also utilized in cell-fate decisions; a well-known example is the bistability of the MAPK cascade which regulates maturation of the *Xenopus* oocyte [32]. While the roles of small motifs have been the focus of research of such systems, the importance of cooperativity among many genes has been stressed theoretically [33].

In this study, we constructed a reference ensemble using McMC and compared it with the results of ES. We extended the research of Ref. [21] to more general network structures and explored both the enhancement of mutational robustness by evolution and how evolution affects the emergence of bistability.

## Model

Genes encoded by DNA in cells are read by RNA polymerase and transcribed to mRNA, which is then used to assemble proteins. A category of proteins called transcription factors acts as activators or repressors of other genes. Many genes regulate each other in this manner and form a complex network called a GRN. GRNs are used to alter the cell state to adapt to environmental changes or to control the cell cycle or cell differentiation. One of the best-known GRN mathematical models is the Boolean network model proposed by Kauffman, in which each fixed point of the dynamical system is considered to represent a cell state [34–36].

In this study, we considered regulatory relations and ignored the details of gene expression. Such connectionist-type modeling has been widely used in theoretical studies [21, 33, 37–44]. We represented GRNs as directed graphs, with nodes as genes and edges as regulatory interactions. For simplicity, we considered GRNs with one input node and one output node. In contrast to Ref. [21], wherein several restrictions were imposed on the network structure, we allowed any network as long as the number of edges from one node to another was at most one. In the following sections, we have restricted our discussion to networks with $N = 32$ nodes and $K = 80$ edges.

A variable $x_i \in [0, 1]$ that represents the expression level of a gene is assigned to each node, where $i$ indicates the node number. $x_i$ obeys the following discrete-time dynamics [38, 39]:

$$x_i(t + 1) = R\left( I\delta_{i,0} + \sum_j J_{ij}x_j(t) \right), \tag{1}$$

where $t$ denotes the time step. $J_{ij}$ represents regulation from the $j$-th node to the $i$-th node. For simplicity, we assume that $J_{ij}$ takes one of the three values $-0, \pm 1$; $+1$ indicates activation, $-1$ indicates repression, and 0 indicates the absence of regulation. The 0th node is the input gene, and $I \in [0, 1]$ is the strength of the input signal [44]. The response of the genes is given by the following sigmoidal function:

$$R(x) = \frac{1}{1 + e^{-\beta(x-\mu)}}, \tag{2}$$

where $\beta$ represents the steepness of the function, and $\mu$ is the threshold. This function is widely used in theoretical studies [33, 40, 44–46]. In the present study, we set $\beta = 2$ and $\mu = 0$. These parameters are the same as those in the neural network model used by Hopfield and Tank [47], and provide a gradual increase to the response function, which reflects the stochastic nature of gene expression [33]. The response function $R(x)$ with $\mu = 0$ is not ultrasensitive because a single gene with this response function has a single fixed point even when the auto-activation loop is attached. Therefore, for the emergence of multistability, many genes must act cooperatively. An example of the parameters of $R(x)$ that a single gene with an auto-activation loop exhibits bistability is $\beta = 6$ and $\mu = 0.5$. Although spontaneous expression $R(0) = 0.5$ is rather high, we do not believe that it caused any problems in the present investigation.

Following the definition of fitness presented by Ref. [21], we set the fitness such that it was larger when the difference in the expression levels of the output gene between $I = 0$ and 1 ("off" and "on") was large. $\bar{x}_{out}(I)$ was the fixed-point value of the output node for fixed $I$; as the initial condition, we set the expression of all genes as 0.5, the spontaneous expression. If $x_{out}(t)$ behaved oscillatorily over time instead of reaching the fixed point, we used the temporal average for $\bar{x}_{out}(I)$; however, the system reached a fixed point in most cases. Fitness $f$ was defined as follows:

$$f = |\bar{x}_{out}(0) - \bar{x}_{out}(1)|. \tag{3}$$

Fitness takes a value in [0, 1] by definition.

We sampled the GRNs using McMC to construct the reference ensemble. For this purpose, we divided the entire range of fitness into 100 bins. McMC enabled us to sample GRNs such that the numbers of GRNs in all bins were almost the same. The GRNs within each bin were randomly sampled in principle. Hereafter, we call this method "random sampling". We performed two types of ES: in one, half of the GRNs in the population were preserved at each generation, and in the other type, 90% of the GRNs were preserved. We referred to these as Evo50 and Evo90, respectively. In the following sections, we mainly describe the results of Evo50,

unless otherwise stated. For each ES, we prepared a random initial population comprising of a thousand GRNs. The details of the computational methods are summarized in the Methods section.

## Results

### Genotypic entropy and speed of evolution

The blue line in Fig 1a shows the base 10 logarithm of the appearance probability $\Omega(f)$ for each bin of fitness $f$ obtained by random sampling. As the logarithm of probability is entropy, we refer it to as the "genotypic entropy." The sum of $\Omega(f)$ was normalized to 1. The probability for $f \geq 0.99$ was $\sim 10^{-16}$. As we could count the total possible number of GRNs as $\binom{N^2}{K} 2^K \simeq 10^{145}$, highly fit GRNs were numerous but rare. The fitness dependence of genotypic entropy is divided roughly into three regions. The majority of GRNs concentrate near $f \sim 0$. The number of GRNs then decreases exponentially with $f$, and, for a very high $f$, they decrease faster than the exponential rate. For comparison, we have shown the genotypic entropy for a steeper response function, $\beta = 4$ and $\mu = 0$, in Fig 1b. It also comprises three regions, namely, the majority around $f \sim 0$, an exponential decrease, and a faster-than-exponential decrease, as in the case of $\beta = 2$ (the jaggy outline is not because of statistical error).

The orange lines represent the average fitness of each generation obtained with Evo50. The vertical axis represents the generation in the downward direction. Evolution progresses almost linearly in the early stage and slows down drastically for a large $f$. The values of $f$ for which the

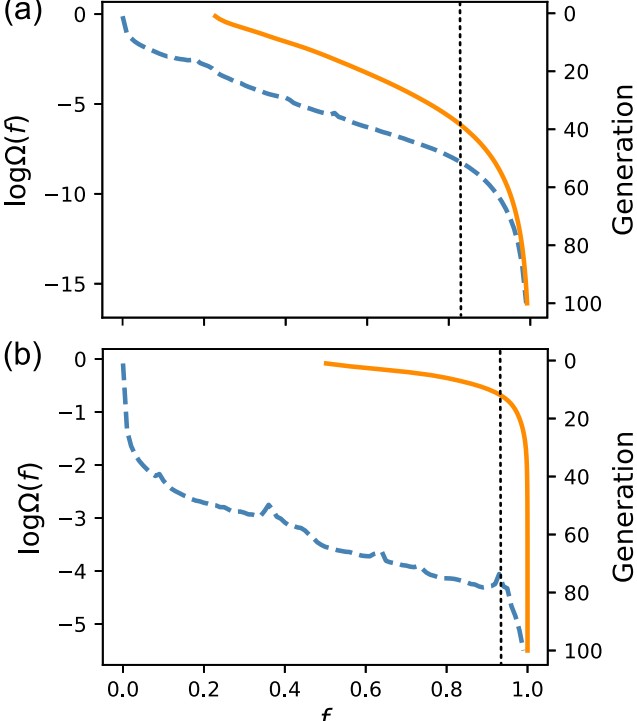

**Fig 1. Genotypic entropy and evolution of fitness.** The fitness $f$ is divided into 100 bins. The blue dashed lines (left axis) show the base 10 logarithms of the appearance probability $\Omega(f)$ of each bin, obtained by random sampling. The orange solid lines (right axis) represent the average fitness of each generation calculated for the lineages obtained using Evo50. Averages were taken over 100,000 lineages. The vertical lines indicate the fitness at which $\Omega(f)$ starts to decrease faster than the exponential rate. (a) $\beta = 2$ and $\mu = 0$. (b) $\beta = 4$ and $\mu = 0$.

increase in fitness starts to slow down roughly coincides with the values for which the faster-than-exponential decrease in $\Omega(f)$ begins for both $\beta = 2$ and 4. This may be because the number of possible destinations that a GRN can transit to by the mutation is restricted by $\Omega(f)$. In other words, when the number of GRNs with higher fitness levels decreases drastically, the possibility that the fitness increases by chance also decreases. A comparison of Evo50 and Evo90 for $\beta = 2$ is given in S1 Fig. Although the evolution was slower for Evo90, the overall tendency was similar. This result suggests that evolutionary speed depends in a large part on genotypic entropy.

## Evolutionary enhancement of mutational robustness

To discuss mutational robustness, we introduce a measure of robustness. In Ref. [21], a single-edge deletion was considered as a mutation. It was found that the edges split into two classes, neutral and essential, for highly fit GRNs, and that the essential edges were minor among them; the essential edge means that the deletion of such an edge leads to fitness close to zero. We considered a single-edge deletion as a mutation in the present model as well, and obtained similar results. We then define the following quantity, $r$, as the robustness measure:

$$r \equiv \frac{1}{K} \sum_{i=1}^{K} f_i',$$

(4)

where $f_i'$ is the fitness after the $i$th edge is deleted, and the sum is taken for all edges. Because $r$ should increase with $f$, comparing the $r$ of GRNs with different $f$ is not meaningful. However, by comparing this quantity for GRNs obtained by random sampling and ES at the same $f$, we can investigate how evolution affects mutational robustness.

Fig 2 shows the average of $r$ against $f$ for random sampling, Evo50, and Evo90. For the ES, we classified GRNs in the obtained lineages by fitness into 100 bins, as with random sampling. The average was taken over all the GRNs in the corresponding bin. For the highest fitness of the ES, only data in the range [0.990, 0.991) were used. $\langle r \rangle$ obtained by ES increased monotonically and coincided with those obtained by random sampling up to $f \simeq 0.5$. For a larger $f$, the value of evolution departed upward from that of random sampling, and the difference became increasingly significant as $f$ increased. Evo50 and Evo90 behaved almost similarly, except for the highest fitness, where Evo90 exhibited a slight decrease. The reason for this decrease is

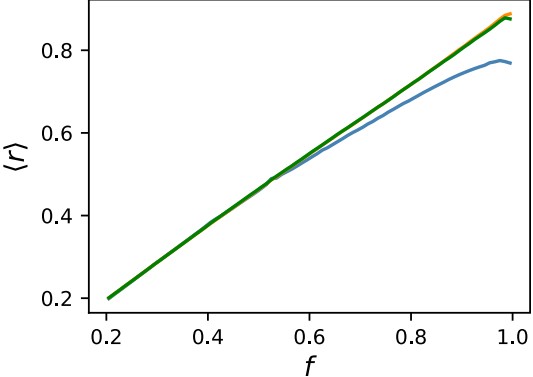

**Fig 2. Average of the robustness measure $r$ against fitness.** The average was taken over all the samples in each bin. The blue line represents random sampling. The orange and green lines represent Evo50 and Evo90, respectively. GRNs obtained by ES were classified according to $f$ into 100 bins, similar to random sampling. The slight decrease at the highest fitness for Evo90 was not caused by a statistical error.

not clear, but the standard errors are very small, and this decrease is not caused by a statistical error.

To scrutinize the difference, we show the probability distributions of $r$ for $f \in [0.5, 0.51)$, $[0.8, 0.81)$, and $[0.99, 1.0]$ in Fig 3a–3c. The data for the ES are taken from Evo50. Considering that the distribution of $f$ within each bin differs for random sampling and ES, we divided each bin into ten sub-bins and reweighed the distribution obtained by ES, so that the distribution of $f$ coincided with that of random sampling. While both distributions roughly agreed for $f \in [0.5, 0.51)$, we observed a deviation for $f \in [0.8, 0.81)$. The two distributions exhibited distinct differences for $f \geq 0.99$, and the distribution obtained by ES was biased to a large $r$ compared with that of random sampling. Therefore, evolution was found to enhance mutational robustness.

What caused this difference in distribution? As stated previously, the edges of randomly sampled GRNs for $f \geq 0.99$ are split into two classes: neutral and essential. Thus, when we delete one edge, $f'$ is either close to $f$ or almost zero, and other types of edges are scarce. Therefore, we investigated the number distribution of essential edges for $f \geq 0.99$ by stating that an edge is essential if $f' < 0.8$. Although this definition is arbitrary, it hardly affects the results. Fig 4 shows the probability distribution of the number of essential edges $n_e$. GRNs obtained by ES

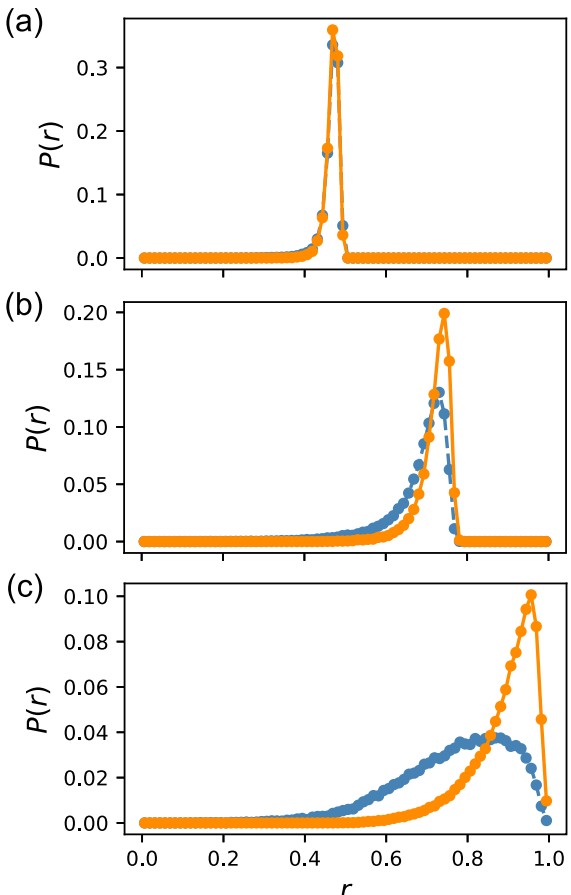

**Fig 3. Probability distributions of the robustness measure $r$.** (a)$f \in [0.5, 0.51)$ (b)$f \in [0.8, 0.81)$ (c)$f \in [0.99, 1.0]$. The orange solid lines represent Evo50 and the blue dashed lines represent random sampling. $r$ was divided into 80 bins because otherwise, unnecessary oscillation appeared in the distribution, owing to the discreteness of the number of essential edges. The distributions for the ES were corrected using the reweighting method described in the text.

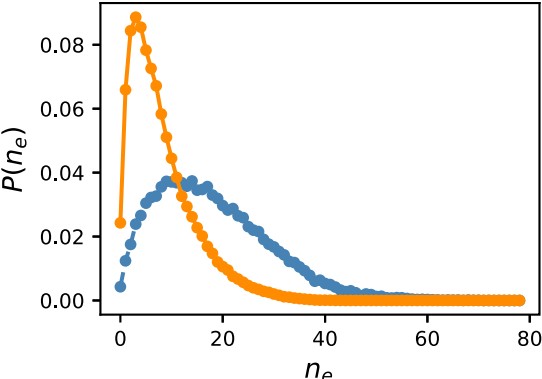

**Fig 4. Probability distribution of the number of essential edges $n_e$.** The data for $f \in [0.99, 1.0]$ are shown. An edge is regarded as essential if the fitness $f$ becomes less than 0.8 after the edge is deleted. The orange solid line represents Evo50, and the blue dashed line represents random sampling. The distribution of the ES was corrected using the reweighting method described in the text.

are significantly biased toward a small number of sides compared to random sampling. The highest probability for ES was at $n_e = 3$, and the distribution was narrow; further, more than 2% of GRNs had no essential edge. In contrast, the highest probability for random sampling was at $n_e = 15$, and the distribution was much broader. The number of GRNs lacking an essential edge was only 0.4%. Therefore, the small number of essential edges is the cause of enhanced mutational robustness by evolution.

One possible explanation for this is that fewer nodes affect the output in the evolutionarily obtained GRNs. To check this, we counted the number of nodes $n_N$ that had at least one path to the output node. Fig 5 shows the distributions, and the distribution of random networks is also plotted for comparison. The peak of the distribution was at $n_N = 28$, which is slightly less than that for the peak of random networks; however, the distributions were indistinguishable between ES and random sampling. Therefore, the number of effective nodes does not cause a difference in the number of essential edges.

## Delayed emergence of bistability

The model in Ref. [21] exhibits bistability as $f$ approaches its maximum value. In other words, when $I$ is changed continuously, two saddle-node bifurcations occur, in contrast to the case of a small $f$, wherein a single fixed point moves to follow the change in $I$. We call the latter type of GRNs monostable. We found that the present model behaves similarly despite the fact that the network structures are significantly different owing to the different restrictions imposed on network construction.

Bistable GRNs are classified into three categories. The first is the toggle switch, in which two saddle-node bifurcations occur within $I \in [0, 1]$. The toggle switch is found, for instance, in phage λ and utilized for adaptation to environmental change [25–27]. The second is the one-way switch [48]. In this case, only one saddle-node bifurcation point is found in the range $I \in [0, 1]$, and another bifurcation point is present outside this range. One-way switches give rise to cell maturation or cell differentiation; a typical example of such switches is the MAPK cascade in the maturation of *Xenopus* oocytes [32]. In the last category, both bifurcation points are outside the range of $I$. As this type may not work as a switch as long as the effect of noise is not considered, we called it "unswitchable." In this study, we did not distinguish among these three and treated them equally as bistable GRNs. It is straightforward to change the definition of bistability to deal only with, for example, toggle switches.

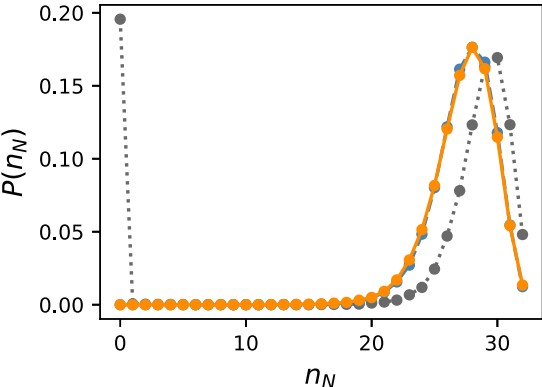

**Fig 5. Distribution of the number of effective nodes $n_N$.** The data for $f \in [0.99, 1.0]$ are shown. If a node has at least one path to the output node, the node is regarded as "effective." The orange solid line represents Evo50, the blue dashed line represents random sampling, and the grey dotted line represents random networks as a reference. The distribution for the ES was corrected using the reweighting method described in the text.

First, we investigated bistability using a strict criterion. Bistability was checked as follows: Starting from the steady state at $I = 0$, $I$ was increased by 0.001, and the dynamics were run until the steady state was reached. This procedure was repeated for up to $I = 1$. Then, the inverse process, from $I = 1$ to 0, was performed. If a difference in $\bar{x}_{out}$ larger than $10^{-6}$ was observed between these two processes in a range of $I$, the GRN was considered to be bistable. We employed such a strict criterion because the monostable GRNs and bistable GRNs were mathematically different as dynamical systems, in that, the number of fixed points was different. Thus, classifying them as strictly as possible is meaningful. In this respect, we may regard monostability and bistability as different phenotypes.

Fig 6a shows the fraction of bistable GRNs $P_2(f)$ against fitness. The blue line is the result of random sampling. The bistable GRNs began to appear at $f \simeq 0.5$ and increased rapidly until all GRNs became bistable for $f \rightarrow 1$. Therefore, a new phenotype of bistability emerged as fitness increased, and all GRNs converged to such a phenotype as fitness approached its maximum value. The orange and green lines are the results of Evo50 and Evo90, respectively. For all data, the standard errors were smaller than the mark. Fig 6a shows that $P_2(f)$ of the ES was substantially lower compared to that of random sampling at the same $f$. In other words, evolution delays the rapid increase in $P_2(f)$. Nonetheless, the eventual emergence of a bistable phenotype is inevitable as fitness increases, because all GRNs are bistable for $f \rightarrow 1$. Evo50 and Evo90 behave similarly except for the early stage of increase; Evo90 initially coincides with random sampling and soon becomes confluent with Evo50. This indicates that the emergence of bistability in evolution depends, partly, on the evolutionary speed.

Although the strict criterion of bistability employed above is mathematically meaningful, very weak bistability may be biologically irrelevant, because it may not be distinguishable from monostability in living systems. Thus, we also investigated bistability using a looser criterion; the checking interval of $I$ was set at 0.01, and the criterion of bistability was set as a difference in $\bar{x}_{out}$ larger than 0.5. The results are shown in Fig 6b. While the rapid increase in the bistable GRNs moves to a higher $f$ compared to that in Fig 6a, the tendency that the emergence of bistability is delayed in evolution compared to that of random sampling remains unchanged. Therefore, we consider that the delayed emergence of bistability is a biologically relevant phenomenon.

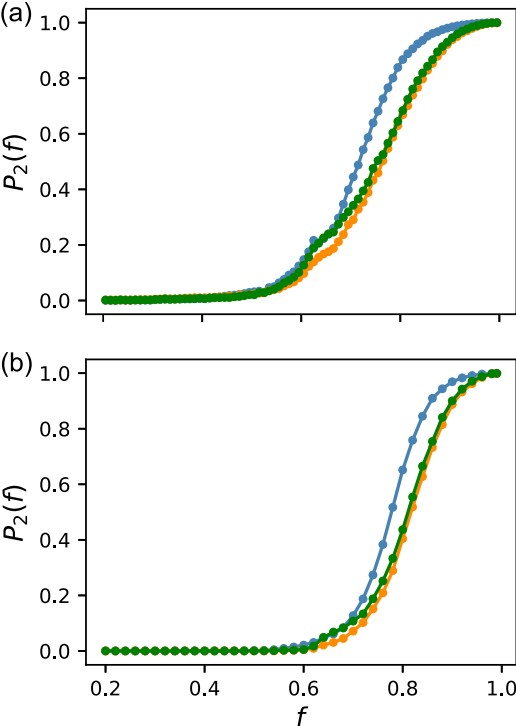

**Fig 6. Fitness dependence of the fraction of bistable GRNs.** The blue line represents random sampling. The orange and green lines represent Evo50 and Evo90, respectively. The standard errors are smaller than the mark. Bistability was checked as follows. First, starting from the steady state at $I = 0$, $I$ increased by $\Delta I$, and the dynamics were run until a steady state was reached. This procedure was repeated for up to $I = 1$. Then, the inverse process, from $I = 1$ to 0, was performed. If the difference in $\bar{x}_{out}$ larger than the threshold $x_{th}$ was observed between these two processes in a range of $I$, the GRN was regarded as bistable. GRNs obtained by ES were classified according to $f$ into 100 bins, similar to random sampling. (a) Strict criterion, $\Delta I = 0.001$ and $x_{th} = 10^{-6}$. (b) Loose criterion, $\Delta I = 0.01$ and $x_{th} = 0.5$.

### Relation between bistability and mutational robustness

So far, we observed the enhancement of mutational robustness and the delayed emergence of bistability by evolution. Thus, we expected that there would be a relationship between them. Fig 7a and 7b show the number distributions of fitness $f$ and robustness $r$ for the GRNs obtained by random sampling for the monostable and bistable GRNs, respectively. Two distributions show pronounced difference. For monostable GRNs, $r$ increases with $f$ and is distributed within a narrow range. In contrast, bistable GRNs include those with very low robustness, irrespective of their fitness values. As a result, the robustness of bistable GRNs is widespread. This difference suggests that if relatively robust GRNs for mutation are favored by evolution, bistable GRNs tend to be avoided.

### Motif analysis

Next, we investigated the network motifs. In the model presented in Ref. [21], coherent feed-forward loops (FFL+), and positive feedback loops (FBL+) were significantly abundant in highly fit GRNs. As the present model allows both auto- and mutual-regulations, we also explored patterns other than triangular patterns. We counted the number of auto-regulations, mutual regulations between node pairs, triangles, and mutual activation or mutual repression of two nodes accompanied by auto-activation of both nodes. Because the motifs are defined as

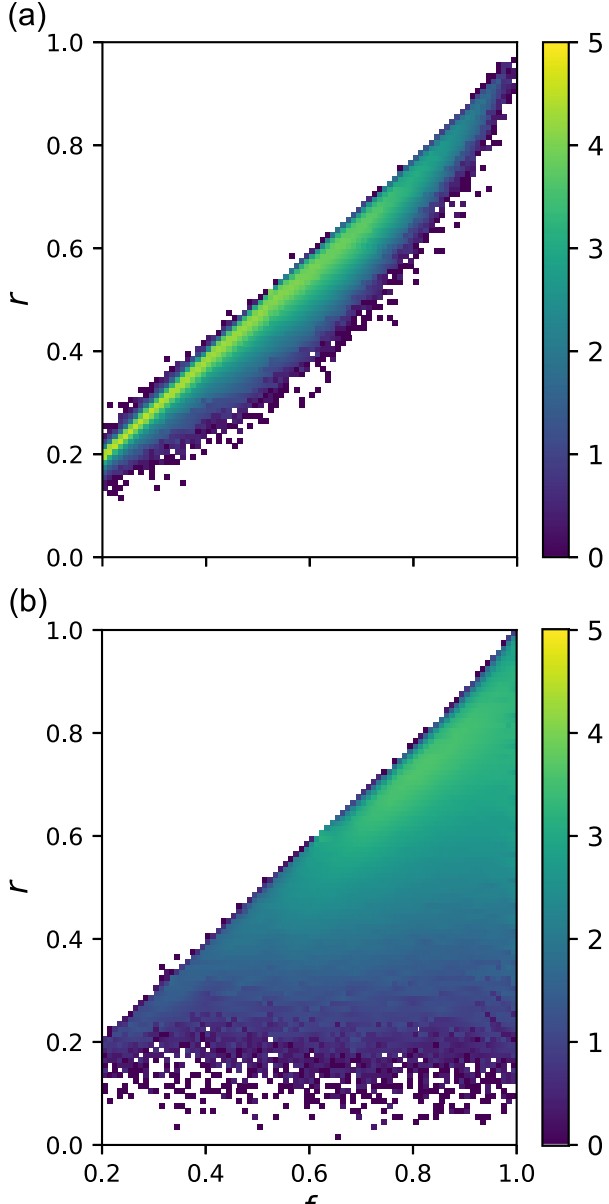

**Fig 7. Number distribution of fitness _f_ and robustness measure _r_ obtained via random sampling.** (a) the monostable GRNs (b) the bistable GRNs. Both _f_ and _r_ are divided into bins of width 0.01. Color of each bin indicates the base 10 logarithm of the number of GRNs. No GRN was found in white bins. The strict criterion was used to determine bistability. Note that the total number of GRNs in each bin of _f_ is about 50000 irrespective of the value of _f_.

network patterns that are abundant compared to random networks [24, 49], we also counted them for random networks.

As a result, the following patterns were greater in number than those in the random networks: auto-activation, mutual activation, mutual repression, FFL+, FBL+, mutual activation accompanied by auto-activations of both nodes, and mutual repression accompanied by auto-activation of both nodes. Although their abundances were not remarkable, we called them motifs. The number distribution of auto-activation is shown in S2a Fig, and those of the other motifs are shown in S3 Fig. Other patterns, such as auto-repression, incoherent feedforward

loop, and negative feedback loop, were fewer in number compared to those in random networks. In S2 Fig, we compare the number distributions of auto-activation and auto-repression; the former is a motif, whereas the latter is not. Although the number of auto-regulations is low, there are very few GRNs that do not undergo auto-activation. In contrast, almost half of the GRNs do not exhibit auto-repression. Thus, auto-activation is favored over auto-repression, but is not indispensable.

Overall, the distributions of these motifs were almost the same for both ES and random sampling. Therefore, whether or not these highly fit GRNs are products of evolution is not reflected in the distributions of these local motifs.

## Path distribution

As a characteristic of the global structure, we counted the number of paths $n_{path}$ connecting the input and output nodes. Fig 8a shows the distribution of paths starting from the input node and reaching the output node without passing the same node more than once. The data for $f \in [0.99, 1.0]$ are shown for random sampling whereas the data for $f \in [0.99, 0.991)$ are shown for ES. Two distributions exhibited a distinct difference; while the probability of GRNs with only one path reached 4% for random sampling, it was 0.1% for ES. In addition, evolutionarily obtained GRNs with less than approximately 100 paths were fewer than those obtained through random sampling. This suggests that the global structures of GRNs obtained

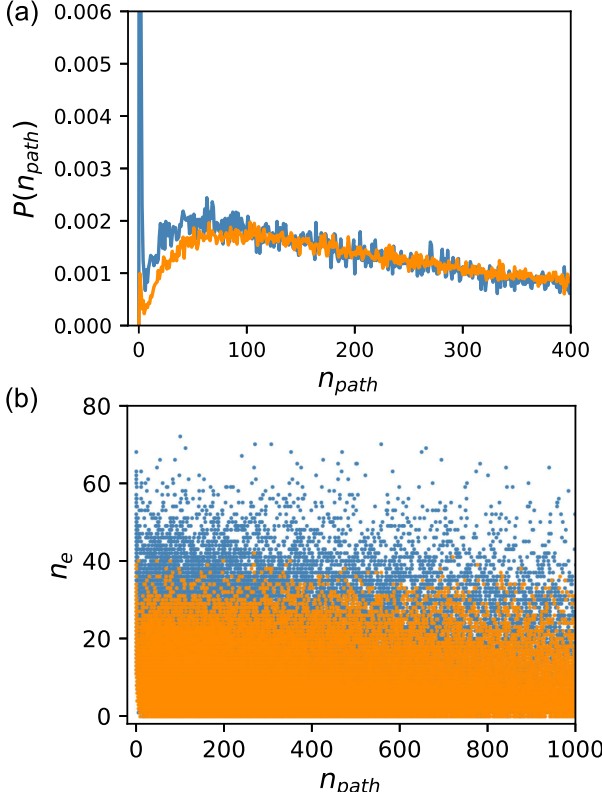

**Fig 8. Number of paths $n_{path}$ starting from the input node and reaching the output node without passing the same node twice.** Orange indicates ES, and blue indicates random sampling. (a) Probability distribution of $n_{path}$. The data for $n_{path} \leq 400$ are shown. The probability of GRNs with only one path reached 4% for random sampling. (b) Scatter plot of $n_{path}$ and the number of essential edges $n_e$. The data for $n_{path} \leq 1000$ are shown. The data for $f \in [0.99, 1.0]$ are shown for random sampling whereas the data for $f \in [0.99, 0.991)$ are shown for ES.

using these two methods differ significantly. However, the difference in path distribution does not explain everything. Fig 8b shows a scatter plot of the number of paths and the number of essential edges. The figure shows that the number of essential edges is lower for ES, irrespective of the number of paths. Even for $n_{path} = 1$, the number of essential edges is distributed broadly. This means that the locations of the essential edges were not limited to the "on-path" locations between the input and output nodes.

### Steady-state evolution of the mutational robustness

Evolutionary simulations would eventually reach a steady state, and the robustness distribution in the steady state was expected to differ from that of random sampling, considering the results shown in Fig 3. To investigate the steady state, we conducted very long simulations of Evo90, and found that the fitness of all preserved GRNs exceeded 0.9999999 at the two-millionth generation. Such extremely high fitness values should be an artifact of the deterministic nature of GRN dynamics and should not be considered biologically relevant. We thus expect that noise is important for high-fitness GRNs.

Instead of introducing noise, we imposed the upper limit $f \le 0.99$ on fitness and conducted simulations of Evo90. In the simulations, fitness values greater than 0.99, were regarded as 0.99. Fig 9a shows the distributions of the number of essential edges $n_e$ for GRNs of $f = 0.99$ at the generation where the fitness of all preserved GRNs first reached 0.99. The results of ten independent runs are shown. The number distributions differed from run to run, and while

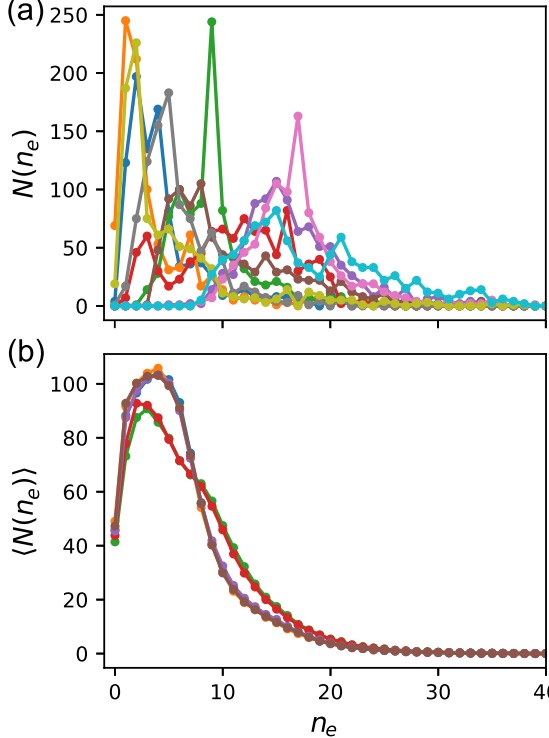

**Fig 9. Distribution of the number of essential edges $n_e$ for GRNs with $f = 0.99$.** We introduced the upper limit of 0.99 for fitness, and conducted Evo90. In other words, a fitness larger than 0.99 was regarded as $f = 0.99$. The data for all GRNs with $f = 0.99$ for each population were used. (a) Distribution of $n_e$ at the generation where the fitness of all the preserved GRNs first reached 0.99. The results of ten independent runs are shown. (b) Average distribution of $n_e$ in the steady state. After 2000 generations, we collected GRNs at every 2000 generations up to one million generations and took their average. The results of six independent runs are shown.

some populations exhibited a small number of essential edges, the overall tendency was that $n_e$ was distributed broadly. As all preserved GRNs have the same $f$, the fitness-driven evolution should have ceased at the generations shown in the figure, after which neutral evolution would continue.

We found that the distribution became steady after approximately 2000 generations. We then conducted runs of one million generations and collected all the GRNs at every 2000 generations. Fig 9b shows the average distributions of $n_e$ for GRNs with $f = 0.99$. The results of six independent runs are shown. Only two distinct distributions were observed; six runs were classified into four and two runs. We considered populations with the same essential edge distribution to be genetically similar. These distributions are biased to low $n_e$ compared to Fig 9a, and the peaks of $n_e$ are 2 and 4 for the two distributions. Moreover, the ratio of GRNs without an essential edge reached approximately 4% of each population. These results indicate that after the fitness distribution reached the maximum, the selection driven by mutational robustness progressed until a steady state was reached. As a result, only a limited number of genotypic groups remained in the steady state; in these six runs, we observed that they converged to only two distinct groups.

## Summary and discussions

In this paper, we proposed a new computational method for studying the properties of evolutionary processes. By generating a reference ensemble via random sampling using the multicanonical Monte Carlo method and comparing it with the outcomes of evolutionary simulations, we can quantitatively explore the commonly observed properties and particularities of evolution. This method is both powerful and general. Using this method, we investigated the evolution of a gene regulatory network model focusing on mutational robustness and bistability.

We found that mutational robustness was markedly enhanced as fitness increased, compared to that in the randomly sampled ensemble with the same fitness, even though the selection is imposed only on fitness. The mechanism for this enhancement can be explained by "second-order selection" [2]. The mutation we considered comprises two successive procedures. First, a randomly selected edge is deleted. Next, a new edge is added between a randomly selected node pair. For fitness values higher than some intermediate values, the edges start to be divided roughly into two types, as observed in Ref. [21]: almost neutral ones, and those causing substantially decreased fitness when deleted. Here, we name the latter type as "essential." If the deleted edge is essential the possibility of fitness to recover via the random addition of a new edge is very low. Therefore, the more essential edges a GRN has, the harder it is for its copy to survive.

The condition for mutational robustness to evolve has been discussed theoretically in the case of neutral evolution, based on population dynamics. According to this theory, the product of population size $P$, the number of edges $K$, and mutation rate $\mu$ should be sufficiently large [2, 50]. As $\mu$ is comparatively large in our ES, we consider that this condition is satisfied. We note a difference: the evolution was not neutral in our ES. The present results showed that mutational robustness was enhanced with increasing fitness in evolution. As mutations that increase fitness are considered rare, fitness increases intermittently. In contrast, mutational robustness can evolve even when mutations are almost neutral because deleterious mutations are expelled by selection.

Based on a GRN model with two-valued fitness, Ciliberti *et al.* reported that mutational robustness is enhanced significantly in GRNs that experienced natural selection compared to that in GRNs randomly selected from viable ones [8]. Several different selection pressures have

also been reported to result in to mutationally robust GRNs [51]. Our results were consistent with these findings.

Next, we discuss bistability. Random sampling revealed that almost all (probably all) GRNs become bistable as fitness approaches its maximum value. Thus, bistability is an inevitable consequence of increased fitness, irrespective of the evolutionary pathway. We confirmed this using an ES. As bistability is not explicitly considered in fitness, it is an emergent property. We may thus regard bistability as a "new phenotype" and the present results indicate that this new phenotype would always appear even if evolution was rewound and restarted. Nagata and Kikuchi obtained the same results for their GRN model [21]. In contrast to our model, auto-regulation and mutual regulation were prohibited in their model. As a result, the distributions of network motifs were different for the two models. Nevertheless, both models exhibited similar bistabilities. Thus, bistability is a common property, irrespective of the network structure. This observation suggests that the possible phenotype is constrained by the form of fitness function.

By comparing the outcomes of ES with random sampling, we found that the appearance of bistability was significantly delayed in evolution. Random sampling revealed that the bistable group of GRNs contained many mutationally fragile GRNs compared to those in the mono-stable group of GRNs. In other words, bistable GRNs and monostable GRNs behave differently in terms of mutational robustness. This is a nontrivial finding that our methodology made possible. This result suggests that the delay in the emergence of bistability may arise from the tendency that mutationally robust GRNs are selected by evolution. Whether this scenario applies to other phenotypes when different fitness functions are considered is of interest for future studies.

A set of GRNs with high fitness composes the neutral space. Thus, we collected members of the neutral space using our method of random sampling by McMC. Ciliberti *et al.* analyzed the structure of the neutral space for the above-mentioned model in which simple random sampling was valid and found that high-fitness GRNs belong to a large neutral space [8]. Unfortunately, a similar analysis was difficult for GRNs obtained using McMC. Instead, we set the maximum value $f = 0.99$ and regarded all fitness values greater than this as 0.99 for conducting long ES. After all the preserved GRNs reached this maximum value, neutral evolution continued and eventually, a steady state was reached. By investigating the steady state, we found that the neutral space is divided into only a small number of parts. This is consistent with the results reported in Ref. [8]. The situation is similar to that studied in the population dynamics mentioned above and is consistent with it, given $PK\mu \gg 1$ [50]. A detailed analysis of the neutral space is required in future research.

Next, we discuss network structures. Characteristic motifs were found for the highly fit GRNs. Among them, coherent feedforward loops are frequently observed in actual GRNs [24, 49, 52]. The following structures are also identified as motifs, which are known to exhibit bistability if the response of each gene is ultrasensitive [23, 32, 53–58]: auto-activation, mutual-activation, mutual-repression, and mutual-activation/repression accompanied by auto-activation of both genes. The last motif is widely observed in multistable GRNs [11, 25, 26, 59–61]. However, these are not relevant to mutational robustness. This suggests that mutational robustness is related to the global structure of GRNs. We found that the number of paths connecting the input and output nodes differed between randomly sampled GRNs and evolutionarily obtained ones. Although it is understandable that GRNs with many such paths are robust because of redundancy, this does not fully explain the origin of mutational robustness.

Finally, evolutionary speed was roughly determined by the number of available GRNs or, in other words, "genotypic entropy." This is partly because of our definition of a single-valued fitness function, which can be computed from the dynamics of a given GRN for a predetermined

single task and has the maximum value. This setup is somewhat artificial, and fitness is not a simple function in reality. However, some experimental studies have addressed the situation discussed in this study. For example, Sato *et al*. reported a similar evolutionary study for a protein [62]. They found that evolutionary speed decreased as fitness increased. We consider that the variation in evolutionary speed in their experiment is explained almost fully by the entropic effect. In a natural situation departing from experimental conditions, evolution is not restricted to proceed in only one direction. When evolution in one direction becomes difficult, the direction changes. We consider that the concept of genotypic entropy affecting evolutionary speed can also apply to such a natural situation.

In summary, we have shown using a new computational method that mutational robustness is enhanced during evolution. We have pointed out the possibility that it affects the emergence of a new phenotype of bistability in GRNs. The research method proposed in this paper can be applied widely to evolution-related phenomena, not restricted to the evolution of GRNs. Further, the basic idea of generating a reference ensemble using McMC can be extended to other fields, such as the learning process of machine learning.

## Methods

### Random sampling

Random sampling was realized using the McMC method (more precisely, entropic sampling [63], which is one of the variations of McMC). The details of this method are described in Ref. [21]. We divided fitness into 100 bins and determined the weight for each bin such that the GRNs appeared evenly, using the Wang-Landau method [64, 65]. One McMC update comprised the following two successive processes: deleting a randomly selected edge and adding a new edge to an unlinked node pair that was also chosen randomly. Whether to accept this change was determined using the Metropolis method. One Monte Carlo step (MCS) comprised $K$ such updates, and we recorded $f$ at every MCS. We sampled GRNs at every 20 MCSs to reduce the correlation between samples. We conducted ten independent runs, each consisting of $10^7$ MCSs. Therefore, we obtained an average of 50,000 samples for each bin. Although inter-sample correlations should have remained to some extent, we named this method "random sampling" in this study. The ensemble of these randomly sampled GRNs was considered as the reference ensemble.

### Evolutionary simulation

We conducted two types of evolutionary simulations: Evo50 and Evo90. For both simulations, we prepared an initial population comprising 1000 randomly generated GRNs. The population size remained unchanged during the simulation. In each generation of Evo50, we selected the top 500 GRNs according to the fitness level and made one copy for each. In the case of Evo90, 90% of the population, namely, 900 GRNs, from the highest fitness were selected for preservation, and the remaining 10% of GRNs were discarded in each generation. We randomly selected 100 GRNs from the 900 preserved GRNs and made one copy for each. Then, these copies were subjected to mutation for both simulations. The mutation procedure for each GRN was the same as that in the McMC procedure; an edge was deleted randomly from the network, and a new edge was then added between a randomly selected unlinked node pair. As these procedures were repeated, the average fitness of the population increased. After 150 generations for Evo50 and 200 generations for Evo90, we selected the GRN with the highest fitness in the population and traced its ancestors to obtain a single lineage. We repeated this evolutionary simulation 100,000 times and 55,000 times independently for Evo50 and Evo90, respectively, and, as a result, we collected 100,000 lineages and 55,000 lineages, respectively.

## Supporting information

**S1 Fig. Evolution of fitness for Evo50 and Evo90.** The orange and green lines represent the average fitness of each generation, calculated for the lineages obtained by evolutionary simulations, Evo50 and Evo90, respectively. Averages were taken over 100,000 and 55,000 lineages, respectively. The vertical line indicates the fitness at which $\Omega(f)$ starts to decrease faster than the exponential rate.
(PDF)

**S2 Fig. Probability distribution of the number of auto-regulations.** (a) Number of auto-activations $n_{A+}$. (b) Number of auto-repressions $n_{A-}$. The data for $f \in [0.99, 1.0]$ are shown. The orange solid line and blue dashed line indicate ES and random sampling, respectively. The black dotted line represents the random networks as a reference. The distribution for the ES was corrected using the reweighting method described in the text. Auto-activation is a motif, whereas auto-repression is not a motif.
(PDF)

**S3 Fig. Probability distributions of the number of motifs.** The data for $f \in [0.99, 1.0]$ are shown. The orange solid lines represent Evo50, and the blue dashed lines represent random sampling. The distributions for the random networks are also shown as references with gray dotted lines. The distributions for the ES were corrected using the reweighting method described in the text. (a) Mutual activation (b) Mutual repression (c) Coherent feed-forward loop (d) Positive feedback loop (e) Mutual activation accompanied by auto-activation of both genes (f) Mutual repression accompanied by auto-activation of both genes.
(PDF)

## Acknowledgments

We thank Koich Fujimoto, Masayo Inoue, Katsuyoshi Matsushita, Nobu C. Shirai, and Hajime Yoshino for their fruitful discussions and comments. We also thank Editage for English language editing.

## Author Contributions

**Conceptualization:** Tadamune Kaneko, Macoto Kikuchi.

**Data curation:** Tadamune Kaneko, Macoto Kikuchi.

**Formal analysis:** Tadamune Kaneko, Macoto Kikuchi.

**Investigation:** Tadamune Kaneko, Macoto Kikuchi.

**Methodology:** Tadamune Kaneko, Macoto Kikuchi.

**Project administration:** Macoto Kikuchi.

**Software:** Tadamune Kaneko, Macoto Kikuchi.

**Validation:** Tadamune Kaneko, Macoto Kikuchi.

**Visualization:** Macoto Kikuchi.

**Writing – original draft:** Macoto Kikuchi.

**Writing – review & editing:** Tadamune Kaneko, Macoto Kikuchi.

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
