## [Decision Letter · Decision Letter 0]

30 Mar 2021

Dear Dr. Kikuchi,

Thank you very much for submitting your manuscript "Evolution enhances mutational robustness and suppresses the emergence of a new phenotype" for consideration at PLOS Computational Biology.

As with all papers reviewed by the journal, your manuscript was reviewed by members of the editorial board and by several independent reviewers. In light of the reviews (below this email), we would like to invite the resubmission of a significantly-revised version that takes into account the reviewers' comments.

We cannot make any decision about publication until we have seen the revised manuscript and your response to the reviewers' comments. Your revised manuscript is also likely to be sent to reviewers for further evaluation.

Sincerely,

Alexandre V. Morozov, Ph.D.

Associate Editor

PLOS Computational Biology

Sushmita Roy

Deputy Editor

PLOS Computational Biology

Reviewer's Responses to Questions

**Comments to the Authors:**

Reviewer #1: General overview

The authors consider in silico gene regulatory networks (GRNs) based on discrete time parallel updates of continuous (transcriptional) expression levels along with (i) an associated fitness function and (ii) evolutionary (mutational) dynamics. In that framework they compare properties of GRNs produced within two ensembles: the "uniform" ensemble of all GRNs of a given fitness (referred to random sampling ensemble) and the GRNs resulting from evolutionary dynamics given an initial starting point. The work's main conclusions are as follows. 1: At given fitness, mutational robustness is higher in the second ensemble; 2: the GRNs of high fitness typically exhibit bistability (and thus hysteresis), but less so in the second ensemble, which the authors interpret as a "delayed phenotype" effect; 3: evolution slows down as one reaches high fitness values.

Main recommendations of the referee

1: The author's use of a "random sampling" is both technically a major accomplishment and a useful tool for obtaining insights into what aspects are due to evolutionary dynamics and not just fitness. The authors use a sampling method (a Wang-Landau multicanonical MC) that produces a quantitative characterisation of the "fitness landscape" which allows the authors to refer to what they call "genotypic entropy". That is very good work and I think some of your readers would benefit further if you put a bit more accent on this point. The authors could have used simulations in the evolutionary ensemble to obtain a characterisation of the same type, leading to genotypic entropy in the evolutionary ensemble too. In physics terminology, you want to answer the question "What is the distribution of fitness in that ensemble", obtainable by direct simulation in the steady state. Theoreticians would be interested to compare that distribution to the Boltzmann one (adjusting the temperature to get the peak at the same fitness position), as far as I know one has little theoretical control over that distribution. Note that this requested computation uses the steady state distribution in the evolutionary dynamics, distribution that depends on the way mutations are produced and accepted from one generation to the next. A "noise level" (related to an effective temperature) is simply the fraction of top fitness individuals produced that are passed on to the next generation in your evolutionary simulation; if you accepted more than 500 individuals, the effective temperature would be higher. I recommend you implement this suggestion so that you no longer use non-equilibrium effects produced by transients and have thus a well defined (steady state) ensemble framework. Changing to this way of computing in your paper would be little work (in fact the authors perhaps already have the data for the noise level they used) but such a change would significantly enrichen the theoretical and conceptual messages of the paper. Furthermore I am quite convinced that the 3 main results put forward by the authors would hold in the steady state of the evolutionary dynamics ensemble. Lastly, note that an advantage of using steady states vs transients is that there then is no dependence on generation number. So to conclude this first point, please do the steady state measurements rather than transient ones and implement the "temperature" parameter in the evolutionary dynamics so that you can scan the average fitness, thereby allowing a much more systematic comparison of the two ensembles.

2: The first of the 3 conclusions given by the authors in their abstract, namely that the second ensemble enhances mutational robustness, is completely expected. Indeed, previous studies have shown that the ensemble based on evolutionary dynamics leads to GRNs with greater mutational robustness than random sampling. That property is very simple to justify qualitatively but it also has a deeper mathematical proof in the case of "0-1" fitness landscapes as given in the paper "Neutral evolution of mutational robustness" (PNAS 1999). So your work can be considered to be a generalization of the results from "0-1" fitness landscapes to arbitrary ones. It is a useful contribution but it is also completely expected. The following analogy from quantum mechanics may help you qualitatively justify the result in your case of continuous fitness landscapes: if one considers a square well potential in 1 dimension, the ground state will have low density at the well's border, corresponding to higher mutational robustness in that ground state compared to the mutational robustness in the trial wave function set to a constant in the well.

3: The second of the 3 conclusions given by the authors in their abstract, namely that bistability is delayed in the evolutionary ensemble, is an observation based on the authors' simulations but remains unexplained. Since the authors established it in their "non equilibrium" framework, we don't know if it is due to the transient dynamics or is a feature of the (steady state) evolutionary ensemble. This gives another reason to follow the request I made of using the steady state ensemble in point 1. Probably the result of bistability holds in the steady state ensemble too. Then a possible interpretation and justification for your result would arise if the "bistability" phenotype was anticorrelated with the mutational robustness. (I have to admit that this seems a bit counter intuitive to me but in fact I have very little basis for having any intuition for this subtle point.) Could you provide a scatter plot and test for such a negative correlation? If the correlation did come out to be negative, you would have a justification for the result, thus a more palatable message. Whether my intuition is right or wrong, I think the paper would be much improved if you could provide new material to put the claim on more solid ground and include a justification. Without a good result there via a revision, I fear your work will have little impact.

4: The third of the 3 conclusions given by the authors in their abstract, namely that evolution slows down as one reaches high fitness, is in a sense inevitable or even a tautology: if fitness cannot improve its rate of change goes to zero... Nevertheless it is just a qualitative observation whereas in a research paper one expects quantitative measures. Furthermore, the current approach of the authors which is to consider the dependence of fitness on generation time has 2 major drawbacks: it is based on the outcome of non-equilibrium dynamics and it is logically inevitable since as one approaches the maximum fitness it can no longer increase. If the authors want to keep this point, I recommend they use the same definition of "slowness" as in Monte Carlo via an autocorrelation time. In the steady state, you can measure that quantity. I would then expect it to grow (slower dynamics) as the selection pressure in the population is increased (or as the mean fitness in the population increases). Allowing such an increase requires that the evolutionary dynamics include a "temperature" parameter, for instance the fraction of kept children over the population size.

Minor comments

1: In the introduction, the authors write "Thus, the central question we pose in this study is whether mutational robustness really evolves". Since the stationary probabilities in your reference ensemble and that produced by evolutionary dynamics inevitably are different, there will necessarily be evolution of mutational robustness in your study. I don't think it is good to say your paper is mainly about this point.

2: You write "this response function has a single fixed point even when the auto-activation loop is attached". Please mention that this feature depends on the values of beta and mu (and possibly at which values auto-activation allows for multistability at the single gene level).

3: Since the Jij space is discrete and your response function R has a maximum that is strictly less than 1, there has to be a drop in GRN entropy at f approaching that limit which is less than 1. This effect must be very small since it is not visible in Fig 1. But it also means that the two curves in that figure have to become vertical before f=1, close or not to the point where the dependence on f becomes super exponential.

4: "In other words, the dynamical system reaches different fixed points for I = 0 and 1." You can improve this sentence so that the readers will understand your point better. The fixed points at I=0 and I=1 are different even at small f. What happens at larger f values is that bistability emerges: in a range of intermediate I values, one has 2 fixed points, having respectively low and high outputs. So although individual genes do not have bistability (for the beta and mu parameters you chose), the GRN does.

5: The high fitness GRNs correspond to amplifying the differential input of node 0. An engineer would be able to design a set of Jijs to do that "simply". Are you able to give such a "design" interpretation to your GRNs of high fitness?

6: Out of curiosity, have you understood whether the autoactivation plays any role, i.e. would the results be almost the same without Jii=+1? I assume that all the high fitness GRN have Jii=0 or +1, that is Jii=-1 is "not good" for fitness. Any results there will be welcome.

7: "This implies that the difference between evolution and random sampling should lie in global network structures." I assume you are comparing the GRNs at a given fitness. Then if it is fitness that drives motifs one does not expect much differences between the two ensembles. The motifs should appear as fitness rises, that is probably the relevant correlation, while the "evolutionary dynamics" ensemble is mainly relevant for mutational robustness.

8: In the Discussion, you write "The new GRN phenotype, bistability, inevitably emerges as fitness increases". I would be more careful about the claim, although the observation holds in your system, it may not be "inevitable". For instance if the parameter mu were adjustable for each gene, one could probably just use a multi-layer fan out architecture to amplify the deviation from 0.5 of each gene's input signal and there would be no bistability. As a result, I am not so sure that your interpretation of "delayed" phenotypic change is generic. Specifically, I am not sure that mutational robustness (enhanced under evolution) is antagonistic with having a particular phenotype (here multistability). This connects to my point 3 in "Main recommendations of the referee" where I think you can improve your study to really reach a conclusive result.

Reviewer #2: In this computational study, the authors simulate a simple model of a genetic regulatory network to investigate how the ensemble of genotypes emerging through an evolutionary process is atypical relative to the ensemble of all high-fitness genomes. For this, they capitalize on a method described in a recent paper sharing one of the co-authors, which employed the multicanonical Monte Carlo technique to sample genotypes in a given fitness bin in an evolution-independent way. This approach allows comparing evolved high-fitness genomes to randomly sampled high-fitness genomes. The authors show that the evolved genomes are more mutationally robust than typical.

I would not necessarily call this reported result groundbreaking; it confirms the intuition many people would probably say they have, but it does so in a clear quantitative way, and the technique used here will likely be useful to other studies. (I, for one, was pleased to learn about the capabilities of McMC.) The paper is well-written and, barring a few comments below, clearly explained. I though the introduction could probably do a better job integrating into the context of the existing literature: currently, it focuses rather narrowly on mutational robustness & GRN, but the idea that evolved genomes are atypical in potentially predictable ways recurs in the broader literature, and to me it seems that the authors' approach is relevant to this broader conversation, which might help this work gain broader visibility. Some examples I'm familiar with are Orr, Genetics (2003); Rice Good Desai, Genetics (2015); Tikhonov Kachru Fisher, PNAS (2020); the work of Erik van Nimwegen. I'm not saying these specific references should be added, merely that these papers, or references therein, might inspire the authors to add a broader paragraph to their discussion section, saying where a similar McMC methodology could be useful.

A few more specific comments:

Major:

1. The "model" section is completely silent on the details of how the evolutionary simulations were performed. Was a single lineage simulated at a time, or a population (how large?) How were mutations implemented? (This is mentioned very late in the paper, and only because it helps explain some finding.) The evolutionary procedure must be clearly described in the methods section.

2. "Evolution behaves conservatively" - I don't know what this means. Evolved genomes are atypical; various observables are skewed in one or another direction, and one can invoke intuitive arguments to rationalize these differences if it helps explain the observations. But interpreting this as "evolution behaves conservatively" / "evolution prevents easy phenotypic changes" does not strike me as justified.

3. In the concluding paragraph, the authors say that one of their "three main results" is the "relationship between evolutionary speed and genetic entropy". The value of this "genotypic entropy" discussion is unclear to me. As you approach a fitness peak, the further increase slows down. This is neither new nor surprising, and if the authors choose to dedicate a paragraph of the discussion to this (e.g. lines 301-321), I think it should be made more clear what specific point they are trying to make. As it is, the discussion is rather meandering and strangely singles out one particular study by Sato et al.

4. "If a difference in x_out larger than 10−6 was observed between these two processes in some range of I, the GRN was regarded as bistable." <-- The value 10^-6 seems so incredibly low that I'm wondering if this is a typo. Surely that level of "bistability" would be completely irrelevant for any biological system? I would imagine (or certainly hope) that the findings in the paper should not change if the threshold is set to something more biologically reasonable. Any value here is an arbitrary choice and I do of course understand that this is not meant as a practical model of any real GRN, but still - any value below e.g. 1% seems like a very odd choice. If the findings DO depend on the threshold value, then I would find this concerning and would certainly like to see some discussion of this.

Minor:

Line 113 - I think some reference or mention that this curve was generated by the McMC protocol is necessary also in the text, not just the figure legend.

Line 117 "GRNs decrease exponentially with f"  number of GRNs?

"Fitness landscape is rough" -> Is this referring to the wiggles in the curve? A rough/rugged landscape usually means something quite specific, and this doesn't seem to be the same meaning as here.

Line 120 "it was divided into three regions" -> unclear what this is referring to

Are there any error bars in Fig 2? Otherwise how can we tell that it is substantially different?

Line 178: Increased monotonically

**Have all data underlying the figures and results presented in the manuscript been provided?**

Reviewer #1: Yes

Reviewer #2: Yes

PLOS authors have the option to publish the peer review history of their article (what does this mean?). If published, this will include your full peer review and any attached files.

Reviewer #1: **Yes: **Olivier C. Martin

Reviewer #2: No
---

## [Decision Letter · Decision Letter 1]

23 Sep 2021

Dear Dr. Kikuchi,

Thank you very much for submitting your manuscript "Evolution enhances mutational robustness and suppresses the emergence of a new phenotype" for consideration at PLOS Computational Biology.

As with all papers reviewed by the journal, your manuscript was reviewed by members of the editorial board and by several independent reviewers. In light of the reviews (below this email), we would like to invite the resubmission of a significantly-revised version that takes into account the reviewers' comments.

We cannot make any decision about publication until we have seen the revised manuscript and your response to the reviewers' comments. Your revised manuscript is also likely to be sent to reviewers for further evaluation.

Sincerely,

Alexandre V. Morozov, Ph.D.

Associate Editor

PLOS Computational Biology

Sushmita Roy

Deputy Editor

PLOS Computational Biology

Reviewer's Responses to Questions

**Comments to the Authors:**

Reviewer #1: General overview

The authors have worked hard to address the criticisms of both referees and the revised version is definitely of quality.

Minor recommendations of the referee

(1) In your abstract you write "This implies that the delayed emergence of bistability is a consequence of the mutation-selection mechanism of evolution". Although I am happy you were able to confirm my intuition on this issue, it is not a proof. You should replace "implies" in this sentence by "strongly suggests".

(2) The term "universal" in physics is strong and I recommend you remove it and tone down the "universality" claims you make in this paper.

(3) The term "randomly sampled GRNs" will be too often interpreted by readers as uniform with no constraint on fitness. I think for your purposes you could replace it by "GRNs sampled uniformly at given fitness" and possibly refer to this as your "reference ensemble" of GRNs. (I prefer "reference ensemble" to "reference set" as it is more general and is directly taken from statistical physics.

(4) "E. Coli" -> "E. coli"

(5) You would benefit from having a native English speaker go through the text.

(6) I found the "Summary and discussions" section rather long.

Other comments

To follow up on my previous report, perhaps you did not understand my sugestions concerning temperature. Your use of different fractions of the population to duplicate corresponds to different selection pressures. I was implicitly suggesting to use the more standard framework whereby the fitness of an individual gives a probability of reproduction. Generally fitness is written as

f = exp(g(genotype)) and the probability to reproduce is f up to a factor depending on the population average of f. This standard method corresponds to Fisher-Wright and allows one to keep the population size constant. My suggestion to introduce a temperature just corresponds to changing that rule to

f = exp(g(genotype)/T). A given T produces a steady state ensemble that will be unique, thereby avoiding (at least theoretically) the disconnected components of a neutral network that you find with your approach.

Reviewer #2: Please see attachment

**Have the authors made all data and (if applicable) computational code underlying the findings in their manuscript fully available?**

Reviewer #1: Yes

Reviewer #2: Yes

PLOS authors have the option to publish the peer review history of their article (what does this mean?). If published, this will include your full peer review and any attached files.

Reviewer #1: No

Reviewer #2: No
---

## [Decision Letter · Decision Letter 2]

4 Dec 2021

Dear Dr. Kikuchi,

Thank you very much for submitting your manuscript "Evolution enhances mutational robustness and suppresses the emergence of a new phenotype: A new numerical approach for studying evolution" for consideration at PLOS Computational Biology. As with all papers reviewed by the journal, your manuscript was reviewed by members of the editorial board and by several independent reviewers. The reviewers appreciated the attention to an important topic. Based on the reviews, we are likely to accept this manuscript for publication, providing that you modify the manuscript according to the review recommendations.

Sincerely,

Alexandre V. Morozov, Ph.D.

Associate Editor

PLOS Computational Biology

Sushmita Roy

Deputy Editor

PLOS Computational Biology

[LINK]

Reviewer's Responses to Questions

**Comments to the Authors:**

Reviewer #1: General overview

In my previous report I no longer had major scientific recommendations. In their revision, the authors handled properly my recommendations and they furthermore improved the logical flow of the paper and streamlined it, taking it from a meadering if not branched and dispersed presentation to a coherent linear demonstration.

I also went through their response to the second referee. Part of his/her reservations overlapped with mine. Since this revision is now readable and follows a clear logic, in my opinion the authors have also responded satisfactorily to the recommendations of this second referee, though clearly he/she had the same reading effort to do as me.

Minor comments

1: In the title and elsewhere, you could replace "numerical" by "computational".

2: I would not use the term "lethal edge": it is the deletion of the edge that is lethal...

3: Figure 7 is a bit misleading because the orange points are displayed after the blue ones and so hide them. It might be better to use crosses for blue rather than the circles so that the overlaps are not a problem.

4: Although Editage was useful, they did not do as good a job on the abstract and introduction (which is overly "wordy"). If you could improve the English there, you'll have more readers going to the end of your paper.

Reviewer #2: I thank the authors for the revisions. They have addressed my concerns. Hopefully the authors find that the changes did indeed benefited clarity.

**Have the authors made all data and (if applicable) computational code underlying the findings in their manuscript fully available?**

Reviewer #1: Yes

Reviewer #2: Yes

PLOS authors have the option to publish the peer review history of their article (what does this mean?). If published, this will include your full peer review and any attached files.

Reviewer #1: **Yes: **Olivier C. Martin

Reviewer #2: No

Figure Files:

Data Requirements:

Reproducibility:

References:

---

## [Editor Report · Decision Letter 3]

27 Dec 2021

Dear Dr. Kikuchi,

We are pleased to inform you that your manuscript 'Evolution enhances mutational robustness and suppresses the emergence of a new phenotype: A new computational approach for studying evolution' has been provisionally accepted for publication in PLOS Computational Biology.

Best regards,

Alexandre V. Morozov, Ph.D.

Associate Editor

PLOS Computational Biology

Sushmita Roy

Deputy Editor

PLOS Computational Biology

---

## [Editor Report · Acceptance letter]

13 Jan 2022

PCOMPBIOL-D-21-00210R3 

Evolution enhances mutational robustness and suppresses the emergence of a new phenotype: A new computational approach for studying evolution

Dear Dr Kikuchi,

I am pleased to inform you that your manuscript has been formally accepted for publication in PLOS Computational Biology. Your manuscript is now with our production department and you will be notified of the publication date in due course.

With kind regards,

Olena Szabo
